# Wear Behavior of Epoxy Resin Reinforced with Ceramic Nano- and Microparticles

**DOI:** 10.3390/polym16070878

**Published:** 2024-03-22

**Authors:** Juana Abenojar, Yolanda Ballesteros, Mohsen Bahrami, Miguel Angel Martínez, Juan Carlos del Real

**Affiliations:** 1Materials Science and Engineering Department, Universidad Carlos III de Madrid, 28911 Leganes, Spain; mbahrami@ing.uc3m.es (M.B.); mamc@ing.uc3m.es (M.A.M.); 2Mechanical Engineering Department, Universidad Pontificia Comillas, 28015 Madrid, Spain; 3Mechanical Engineering Department, Institute for Research in Technology, Universidad Pontificia Comillas, 28015 Madrid, Spain; yballesteros@comillas.edu (Y.B.); delreal@comillas.edu (J.C.d.R.)

**Keywords:** cavitation erosion, wear, nanoparticles, silicon carbide, boron carbide, coating, nanocomposites, epoxy resin

## Abstract

Cavitation erosion poses a significant challenge in fluid systems like hydraulic turbines and ship propellers due to pulsed pressure from collapsing vapor bubbles. To combat this, various materials and surface engineering methods are employed. In this study, nano and micro scale particles of silicon carbide (SiC) or boron carbide (B_4_C) were incorporated as reinforcement at 6% and 12% ratios, owing to their exceptional resistance to abrasive wear and high hardness. Microparticles were incorporated to assess the damage incurred during the tests in comparison to nanoparticles. Wear tests were conducted on both bulk samples and coated aluminum sheets with a 1mm of composite. Additionally, cavitation tests were performed on coated aluminum tips until stability of mass loss was achieved. The results indicated a distinct wear behavior between the coatings and the bulk samples. Overall, wear tended to be higher for the coated samples with nanocomposites than bulk, except for the nano-composite material containing 12% SiC and pure resin. With the coatings, higher percentages of nanometric particles correlated with increased wear. The coefficient of friction remained within the range of 0.4 to 0.5 for the coatings. Regarding the accumulated erosion in the cavitation tests for 100 min, it was observed that for all nanocomposite materials, it was lower than in pure resin. Particularly, the composite with 6% B_4_C was slightly lower than the rest. In addition, the erosion rate was also lower for the composites.

## 1. Introduction

Nowadays, in many systems like hydraulic equipment, fluid pump impellers, ship propellers, hydrodynamic bearings, fluid seals, inlets to heat-exchanger tubes, diesel engine wet cylinder liners, hydrofoils, liquid metal power plants, steam turbines, etc., mechanical erosion could be a great problem. This affects energy production and environmental protection since hydroelectric power is frequently used due to its commonality and significant contribution to total energy production [1]. The degradation of turbine blades caused by erosion processes such as cavitation and sediment erosion produces a challenge and an additional cost associated with energy production [2]. Cavitation is the process of the formation of vapor bubbles in low-pressure regions within a fluid, produced when the hydrostatic pressure acquires a value lower than the saturation vapor pressure [3]. High-speed turbines operating under medium and low head conditions are closed turbines that function under variable pressure. These turbines are particularly susceptible to cavitation, with a specific operational area severely affected. Consequently, these turbines experience decreased performance over time due to significant damage caused by various factors.

As per Bernoulli’s equation, when the pressure in any part of the turbine falls below the evaporation pressure, the liquid boils, forming numerous small vapor bubbles. These bubbles, predominantly formed due to low pressure, are carried by the flow to higher pressure zones where they collapse suddenly as the vapors condense back into liquid [4]. This rapid collapse creates a cavity, causing the surrounding liquid to rush in and fill the void [5].

Although cavitation or erosion studies often utilize both numerical and experimental methods, relying solely on numerical techniques may seem efficient, yet it disregards crucial factors. Numerical methods, while time- and resource-saving, encounter challenges stemming from various parameters [2,6,7]. These parameters interplay, where temperature fluctuations within the liquid flow impact pressure fluctuation amplitudes and wave velocities [8,9]. Hence, an experimental approach becomes essential, considering the myriad parameters influencing the process: pressure, temperature, rotational speed, slurry sizes and shapes, and micro-jet formations during bubble implosion. Consequently, failure mechanisms vary significantly due to these multifaceted parameters [10,11].

A great variety of materials and many different surface engineering processes have been applied to increase the cavitation resistance [12,13]. Many metallic and intermetallic composites have been used as coatings because of their good resistance to cavitation erosion due to their high strain hardening rate, high hardness, good oxidation and corrosion resistance [14,15]. This can be done in many ways, and so, Babu et al. developed a microwave-synthesized Ni-SiC material and thermally spray coated the 316L stainless steel of the turbine [16]. Likewise, many ceramic composites have been used for their excellent wear resistance and high hardness. They are considered prime candidates for applications requiring high resistance to cavitation erosion corrosion, among which are carbides [17,18]. Surface treatments on turbines, such as plasma nitriding, laser hardening and melting are also efficient [19]. Brijkishore et al. [20] compiled different surface treatments to mitigate the extraneous effects created by cavitation in hydraulic turbines. While cavitation erosion is typically undesirable in turbines, there are instances, as demonstrated in the study by Peng Chin et al. [21], where it is deliberately employed. In such cases, micro-abrasives have been shown to enhance ultrasonic cavitation erosion of reservoir rocks, thereby contributing to the overall effectiveness of ultrasonic treatment in oil wells within the oil industry. In addition, hydrodynamic cavitation has other applications, such as treatment of effluent containing pharmaceutical compounds in combination with other techniques [22], or as an aid in the degradation of bisphenol A [23].

In the case of polymer coatings, issues related to adhesion and micro-structural defects have been identified. The latter significantly impacts the initiation of cavitation pitting [24]. Epoxy resins, however, have emerged as promising candidates for coating materials to resist damage from mechanical erosion, whether applied as varnishes or paints [25]. This is due to the excellent resistance to acids, bases and solvents as well as strong mechanical properties [26]. However, epoxy resins are brittle materials, when hard angular particle impacts a brittle material, plastic deformation occurs in the contact area due to the high compressive and shear stresses, and accordingly, a radial/medium crack is formed. The plastic deformation provides large tensile stresses that lead to the initiation of lateral cracks and consequently to the removal of material [27]. The current trend is shifting towards the production of epoxy-elastomer-based composite materials or epoxy/polyurethane mixtures [13,28], initially incorporating micrometric particles, and later incorporating nanometric particles, such as alumina [29,30], silica [29,31] titanium oxide, zinc oxide, clay or magnesium silicate [24] and carbon-based materials [32]. Conversely, epoxy-based varnishes, silicone, or polyurethane-based paints can be applied conveniently and cost-effectively. The micro/nano transition arises as a consequence of the fact that micrometer particles produce a fully brittle model. In contrast, the use of nanometer particles produces a ductile model, varying the depth of the crater and the residue accumulated at its edge. Therefore, the size of the particles used as reinforcement modifies the elimination model [33]. The brittle-ductile transition can also be observed in polymers in unreinforced polymers. This transition between brittle and ductile fracture is the underlying cause of varying erosion rates in polytetrafluoroethylene (PTFE) [34], depending on the specific erosion stages.

The wear phenomena that occur due to external processes and can disrupt production chains include abrasion, friction, erosion, cavitation, and corrosion, among others. Given that cavitation involves hydraulic transport and is a wear phenomenon, it is logical to investigate the type of wear affecting the materials involved in this study. Additionally, considering the significant impact of the high surface/volume ratio of nanoparticles on the epoxy curing process, microparticles have been included for comparison to assess their influence on wear and erosion caused by cavitation, utilizing different sizes of carbide particles such as silicon (SiC) and boron carbide (B_4_C) as reinforcement or filler in an epoxy resin. The composite materials obtained are then compared with pure resin.

Boron carbide and silicon carbide ceramics were chosen due to their exceptional characteristics such as low density, elevated melting point, superior hardness, exceptional resistance to chemical erosion, and elevated durability, among others. The employment of substances with remarkable wear resistance not only prolongs the operational lifespan of tribo-components but also diminishes the production of wear debris [35].

This study is crucial since the available information on the wear and/or cavitation erosion of such engineered nanocomposites is limited. Consequently, the objective of the present study is to investigate the effect of reinforcing epoxy matrices with ceramic nanoparticles of SiC and B_4_C on wear and mechanical erosion caused by cavitation and to compare it with the effects of microparticles.

## 2. Materials and Methods

### 2.1. Materials

The material composition of the samples was as follows: pure epoxy samples derived from EPOFER Ex 401, featuring EPOFER E432 hardener, hereinafter referred to as EPOFER (supplied by Feroca, Madrid, Spain). The main properties are a viscosity of 1300 cps, a curing temperature of 25 °C, and a curing time of 24 h.

Nano-composite samples were created using EPOFER as the matrix, loaded with nanometric particles of SiC (at 6 and 12 wt.%) and B_4_C (at 6 and 12 wt.%). To compare the damage caused in the nanocomposites, microparticle fillers of SiC and B_4_C with sizes of 10 and 7 µm, respectively, were used in the same resin.

These percentages are in accordance with the bibliography on particulate reinforcements [36] and previous work conducted by the research group on epoxy resin with carbides. The nanometric B_4_C was provided by PlasmaChem GmbH (Berlin, Germany), with an average particle size of 30–60 nm, and its morphologies are depicted in Figure 1a. B_4_C particle size of 7 µm (in Figure 1b) was supplied by Strem Chemicals (Bischheim, France). Nanometric SiC was supplied by Bioker Research S.L. (Oviedo, Spain), featuring an average particle size of 80–100 nm (in Figure 1c); SiC particles of 10 µm (in Figure 1d) were supplied by Carburos Navarro S.A. (Cuenca, Spain). Named hereafter E6SCn/m, E6BCn/m, E12SCn/m, and E12BCn/m; where E is referred to Epofer resin, 6 or 12 is the particle percentage, SC and BC are SiC and B_4_C, respectively, and n or m is nano or microparticles.

### 2.2. Sample Preparation

The chemical base of the nano- and microcomposites was prepared by extrusion mixing, with the extruder passage repeated ten times per case. Bulk samples of nano- and microcomposites, and pure resin, were then obtained by molding in silicone molds, following prior deaeration of the resin. The shapes of the samples varied depending on the type of test to be conducted—samples for wear tests differed from those for cavitation tests. There were two sample shapes for wear tests: bulk and a 1 mm coating on an aluminum substrate (Figure 2a); the dimensions of the bulk samples were 25 mm diameter, and 25 mm × 25 mm for the aluminum sheets; five samples by material were prepared. For cavitation tests, samples were prepared as a 1 mm coating on aluminum tips (Figure 2b). The diameter of the tips was 10 mm, and five tips were coated with material. The surfaces of the test samples, both resin and nano- and microcomposites, were initially sanded roughly with 600 μm sandpaper and then finely with 1200 μm sandpaper, in a circular way so as not to create preferential direction. The aim of this sanding was to achieve a uniform thickness across the entire coating surface. Before applying the coating for wear and cavitation testing, the aluminum plates and tips were cleaned with MEK (methyl-ethyl ketone) and chemically treated with 3-glicidoxy-propyltrimethoxysilane (GPTMS) from Sigma Aldrich (Saint Louis, MO, USA). A 1% silane solution was hydrolyzed for one hour in MilliQ water at pH 5, and the tips were then dried for one hour at 100 °C. This treatment ensures good adhesion of the coating to the epoxy resin or nano- and microcomposites to the aluminum surface [31].

### 2.3. Wear Test

Dry wear tests were carried out at room temperature using a pin-on-disk tribometer (Microtest, Madrid, Spain). A 6 mm diameter alumina ball was used for the pin. Test conditions were 120 rpm, with an applied load of 15 N, relative humidity below 30%, and a friction radius of 8 mm. The sliding distance was 1000 m. For the wear test, both the bulk composite and a 1 mm thick coating were used, as shown in Figure 2a.

The wear and coefficient of friction were calculated following the ASTM G99 05 (2010) standard test [37]. Wear was evaluated by volume loss, according to Archard’s equation (Equation (1)). Volume loss is relative to the wear track and pin (Equation (2)), where the wear track radius^®^ and wear track width (d) were measured using a DSX500 opto-digital microscope (OM) supplied by Olympus Corporation (Tokyo, Japan); while (r) represents the pin end radius. In these tests, it is assumed that there is no significant pin wear, and the wear debris is intentionally left on the track since it plays an important role in fretting wear models [38].
(1)$$W=\frac{Volume\;loss}{Load\times Sliding\;distance}(\frac{{mm}^{3}}{N.m})$$
(2)Volume loss (mm3)=2πRr2sin−1d2r−d44r2−121/2

Later, wear tracks were analyzed by scanning electron microscopy (SEM), Philips X-30 model (Philips Electronic Instruments, Mahwah, NJ, USA), to determine the mechanism of wear [39].

### 2.4. Cavitation Erosion Test

The cavitation tests were conducted in accordance with standards ASTM G32-16 (2021) [40] by means of a piezoelectric vibratory apparatus Branson 450D (Branson Ultrasonics, Brookfield, CT, USA). This method produces cavitation damage on the face of a specimen by subjecting it to high-frequency vibration while immersed in a liquid. The vibration induces the formation and collapse of bubbles in the liquid, resulting in damage and erosion (material loss) of the specimen. The vibratory apparatus used for these tests produces axial oscillations of a test specimen inserted to a specified depth in the test liquid; in this experiment, water served as the test liquid. The cavitation surface was evaluated by visual observation (VO) and SEM. Similar technical analyses were utilized [39] to determine the cavitation erosion mechanism of plastics. Before SEM analysis, both wear and cavitation surfaces were coated with gold by a sputtering system, using a Leica EM ACE200 low vacuum coater (from Leica Microsystems S.L., Hospitalet de Llobregat, Spain).

The cavitation test parameters included using distilled water at 25 °C as the temperature of the fluid, with atmospheric pressure maintained at 1 atm, frequency of 20 kHz and amplitude of 25 mm. The duration of the test was 80 min, conducted in intervals of 5 to 10 min. Before weighing to obtain the mass loss over time and the cumulative mass loss, the samples were dried in a desiccator for one hour. The steady-state erosion wear rates, both maximum and terminal, were obtained from the slope of the cumulative mass loss curve. Since the test typically exhibits a maximum followed by decreasing erosion rates, the wear rate was derived from the first part of the curve. Once the maximum erosion wear rate was calculated, its intercept with the x-axis gives the incubation time. The Mean Depth of Erosion (MDE) was obtained using the mass loss, the density and the sample area, as illustrated in Equation (3). The sample area, representing the tip area was 2.778 cm^2^, and the density was calculated with volume and mass of the sample.
(3)$$MDE\;(mm)=\frac{Mass\;loss}{Area\times density}$$

## 3. Results and Discussion

### 3.1. Wear Tests

The friction coefficient (FC) curves versus sliding distance can be divided into two parts. In the first part, the FC increases rapidly, reaching a maximum value, which is typically related to the formation of the track and mainly involves abrasive wear. Depending on the materials, this phase is reached after approximately 100 m of track. In the second part, the FC remains constant, with fluctuations that may be attributed to alternating adhesion-abrasion zones. The average value of this zone is usually reported as the coefficient of friction. However, this zone may not stabilize due to the effect of abrasive particles on the track, known as the third-body effect, which increases abrasion [41].

Figure 3 depicts the typical curves for these materials. In these cases, the FC initially increases in the first meters and reaches a maximum peak. Subsequently, it decreases, followed by small fluctuations, and then gradually increases slightly until the end of the test. A trend line is drawn in this ascending part. The slope of this trend line is small, on the order of 10^−5^ (Figure 3), indicating a minimal ascent, and thus, the FC can be considered as the ordinate at the origin of this trend line. The fluctuations arise from the formation and growth of wedges, caused by the material’s deformation and retention by loose particles, which hold onto debris dust [7]. This debris dust persists throughout the test, resulting in wedgelike agglomerates by the end. The wear mechanism in this case is abrasive-adhesive, which may lead to an increase in the FC, sustaining the wear, or even decrease it.

Figure 4 depicts the friction coefficients (FCs) for all the materials studied, with each value obtained as the average of three specimens. Bulk samples exhibit lower FCs than the coatings on aluminum, except for the 12% nanocomposites (E12SCn and E12BCn). Among the bulk samples, the E12SCn nanocomposite has the highest FC (0.70), while the 6% nanocomposites (E6SCn and E6BCn) have the lowest (0.25 and 0.30, respectively). No major differences are found in the coatings, with all falling between 0.4 and 0.55. However, deviations are observed within each measurement, possibly due to wedgelike agglomerates. The higher value for E12SCn in bulk can be attributed to wedgelike agglomerates that can detach during wear due to the lack of anchoring with the matrix.

The wear caused by the alumina counter material in the pure resin (E) is higher than in the nanocomposites and microcomposites (Figure 5). Surprisingly, the wear of the coating is lower than that of the bulk E samples, contrary to what was observed for FC. This difference can be attributed to the greater deformation in the bulk E samples compared to the coating. In the bulk samples, the pin has greater contact with the resin, leading to more material removal. The effect of the pressure of the counter-material (pin) produces a greater deformation in the massive part than in the coating. Conversely, the thinness of the coating and its adhesive bond to the base metal limit its deformation. As there are no hard particles, the adhesion of the waste powder occurs when passing the pin again. This mechanism is facilitated on the coating due to the support underneath, forming small agglomerates of the same material on the track, which can increase the FC. On the other hand, the deformation and high contact in the pure bulk resin mean that the debris powder produced does not remain on the track, increasing wear, as depicted in Figure 5.

The wear of the nanocomposites in bulk varies with the proportion of added carbides (Figure 5), resulting in higher wear for the composites containing 12% carbides. Interestingly, a significant decrease in wear of 66% and 86% is observed for E6SCn and E6BCn, respectively, in the bulk samples compared to the coatings. For samples containing 12% carbides, wear values are more comparable between the two types of samples. However, higher wear is still predominant in the coating, with 2% and 21% for E12SCn and E12BCn, respectively. The wear mechanism for the nanocomposites seem to be abrasive-adhesive, involving material pull-out and the formation of wedgelike agglomerates. In the case of composites containing 12% carbides, the possible presence of agglomerates of nanoparticles within the composite material, which upon detachment leave more free volume, may also play a role in increased wear.

Considering the angular and pointed shape of the microparticles (Figure 1b,d), it is logical to observe an increase in wear in microcomposites compared to nanocomposites (Figure 5), since abrasive wear is favored. However, the entire mechanism remains abrasive-adhesive. An increase in wear is also evident with the percentage of particles, especially for E12SCm and E12BCm in the case of bulk samples (42% and 51%, respectively), and 26% for E6BCm, which always higher than the corresponding coatings. In contrast, there is little difference observed among the coatings, and they remain similar to those of the nanocomposites in the range of 3–4 × 10^−4^ mm^3^/m.N. The role of aluminum is to favor adhesive wear, contributing to the formation of wedgelike agglomerates (red arrows in Figure 6 and Figure 7).

The wear behavior can be understood through various wear mechanisms, with the interaction between nanoparticles and resin playing a pivotal role. Typically, wear tracks exhibit abrasive-adhesive characteristics, evident in Figure 6, showcasing abrasion lines (blue circles) and wedgelike agglomerates (red arrows). These abrasion lines are notably pronounced in pure resin and nanocomposites containing 12% particles, resulting in elevated friction coefficients (Figure 4). Additionally, small cracks, attributed to fatigue, are observed in both pure resin and nanocomposites with 12% particle content (Figure 6a,a’,e). However, in nanocomposites with 6% filler, adhesive wear is also evident, leading to reduced wear (Figure 5) and friction coefficients (Figure 4). Typically, agglomerates resulting from adhesive wear tend to accumulate more at the edges of the track. However, they are also distributed throughout the track, causing fluctuations in the coefficient of friction (see Figure 3). The center of the track experiences more abrasive wear, leading to its non-flat nature and certain depth. The discrepancy between SiC and B_4_C is attributed to the anchoring of nanoparticles (Figure 6b–e). B_4_C nanoparticles feature hydroxyl groups on their surface originating from the manufacturing process [36], enabling better adhesion to pure resin and minimizing detachment compared to SiC. However, the nanoparticles may act as a third body, intensifying abrasive wear. Fatigue wear arises from cyclic mechanical stress, fostering the formation and propagation of cracks beneath the surface, resulting in structural damage such as transverse and vertical cracks, particularly evident in composite materials.

Figure 7 compares the bulk samples with the coatings, specifically focusing on the composites with micrometer SiC. The micrographs display striking similarities, showcasing areas of buildup indicated by red arrows and more abrasive regions marked with blue circles. Previously, Figure 4 and Figure 5 demonstrated equivalent wear between bulk E6SCm and the coatings of E6SCm and E12SCm. This consistency is further evident in Figure 7a,b,d. However, in the case of the bulk sample E12SC (Figure 7c), areas of reduced adhesion are noticeable, resulting in increased wear (Figure 5) and decreased FC (Figure 4).

### 3.2. Cavitation Erosion Tests

Figure 8 illustrates the mass loss rates observed in cavitation tests conducted on pure resin, as well as nano- and microcomposites. Pure resin (E) exhibits a higher mass loss rate compared to the composites, with a delayed maximum value observed at 25 min. In contrast, nano- and microcomposites display lower mass loss rates than pure resin, ranging from 55% to 88%. Interestingly, all composites reach their maximum peak at the same point, with a delay of 15 min, which is shorter than that of pure resin.

The differences between nano- and microcomposites are more apparent in the cumulative mass loss (Figure 9) compared to Figure 8, since the pure resin curve was omitted. Some minor distinctions emerge in the accumulated mass loss (Figure 9) between nano- and microcomposites, with nanocomposites exhibiting lower accumulated mass loss than microcomposites. Regarding particle size and percentage, there is a clear trend: as the particle size decreases, the cumulative loss diminishes, mirroring the trend with the percentage. The difference between micro and nano in relation to the accumulated mass loss is only a size relationship between the particles. Larger size results in more mass loss. Therefore, for nanocomposite coatings, the cumulative mass loss follows this ascending order: E6BCn, E12BCn, E6SCn, and E12SCn. On the other hand, for the microcomposite coatings, the differences are smaller. The E6SCm and E12SCm samples show higher cumulative mass loss and are overlapping. For E6CBm the mass loss is slightly higher than for E12BCm, contrary to the nanocomposites.

The impact energy is absorbed by the solid material through elastic deformation, plastic deformation, or fracture; the latter two processes result in material erosion. The greater the amount of energy a material can absorb through elastic or plastic deformation, the greater its resistance to cavitation erosion. Erosion is typically linked with surface mass loss and occurs after an incubation period. During this period, materials undergo elastic or plastic deformation [41].

The steady-state erosion wear rates (maximum and terminal) are determined from the slope of the cumulative mass loss curves (Figure 9), following the ASTM G32-16 standard [40]. As the test demonstrates a peak followed by declining erosion rates, the wear rate is derived from the initial segment of the curve in the Figure 10 example. Once the maximum erosion wear rate is determined as the line slope of maximum erosion wear, its intersection with the x-axis, for y = 0, yields the incubation time. The maximum erosion wear rate was determined the same way for all samples, in the part of the cumulative mass loss curve with a higher slope, which is between 10 and 25 min of the test (Figure 10).

Table 1 reveals that the maximum and terminal erosion rates in nanocomposites are lower than those in the pure resin. Additionally, slight differences are observed among nanocomposites. In the case of E6SCn and E612SC, maximum and terminal erosion rates are similar, with their values falling within the measurement error margin of ±0.02. Conversely, in E6BCn, the maximum erosion rate is lower than that in E12BCn. This variation may be attributed to the lower density of E12BCn compared to E6BCn, as porosity significantly influences both incubation time and erosion rate, since cavitation may be accelerated around the porosities [42]. Concerning microcomposite coatings, neither the maximum nor the terminal erosion rates changed significantly compared to nanocomposite coatings, except for E6SCm which is 20% higher than its nano equivalents. This is probably due to further irregularities in this coating due to the occasional presence of poorly adhering surface particles.

The incubation time in Table 1 is slightly higher for pure resin than for the nanocomposite coating. This slightly longer incubation time may be due to a small elastic deformation in the resin, which recovers before the nucleation of cavitation cracks begins. The slight differences found can also be due to measurement error (±1 min). However, for microcomposite coatings, it is considerably lower. Moreover, these times are very low when compared to metals with incubation time in hours [43], but the easy replacement of these coatings against metal repair allows for its industrial use.

The MDE parameter is derived from the mass loss, density, and sample area, as depicted in Equation (3). The sample area measures 2.778 cm^2^, and density is determined by volume and mass calculations. The MDE results obtained after 80 min are detailed in Table 1. These MDE values align with the mass loss data, indicating that the erosion depth is reduced in nanocomposites compared to pure resin, while remaining within an intermediate range for microcomposites. Specifically, within the EBC category, MDE values are lower than those in ESC, with a measurement error of ±5 µm. Notably, the lowest MDE corresponds to E6BCn, possibly due to the superior anchoring of nanoparticles to the epoxy matrix, as previously discussed.

Crazing and cracks joining can be observed at low magnifications in pure resin (E) (Figure 11a). These mechanisms are more prevalent in the center of the specimens (Figure 11b), while the edges of the samples exhibit smooth areas where cavitation is not detected. Upon closer examination, cavitated resin (E) specimens display brittle and root-shaped fractures, as well as delamination patterns (Figure 11c,d). Delamination happens in areas with poor or no bonding between adjacent layers. It occurs by crack extension along a plane parallel to the surface, at the substrate interface. This effect can only happen on the epoxy resin, since it is a brittle fracture and only the form is slightly different, as is the case in root shape fractures or the material’s own brittle fracture (perpendicular to the surface).

The SEM results of cavitated specimens for nanocomposite coatings are illustrated in Figure 12. Similar to pure resin, reduced cavitation is observed at the edges, but the rest of the specimen displays increased cavitation due to small air bubbles, particularly around the edges (Figure 12a,c). Thus, porosity plays a crucial role in these materials, with nanocomposites exhibiting closed porosity ranging between 3 and 5%, and higher levels at 12% in some samples, serving as nucleation sites for crack formation. The closed porosity arises from inadequate deaeration during the manufacturing process and premature skin formation before the nanocomposite is fully cured, as a consequence of increased mixture viscosity. The improved resistance to cavitation in nanocomposites can be attributed to several processes. Firstly, there are more areas exhibiting root-shaped fracture patterns (Figure 12a,c), resulting from fatigue mechanisms (Figure 12a), which are slower than brittle fractures. Additionally, a combination of brittle and root-shaped fractures alongside plastic deformation is observed (Figure 12b–d). Furthermore, a significant amount of ductile fractures facilitates material deformation without fatigue. Consequently, nanocomposites enhance cavitation resistance, leading to decreased cumulative mass loss (Figure 9) and incubation times comparable to pure resin but longer than microcomposites (Table 1). These cavitation mechanisms have been previously described by Correa et al. [24] for epoxy-based composites.

On the contrary, microcomposite coatings exhibit fewer air bubbles, although they closely resemble pure resin (Figure 13). Brittle breaks and delamination are evident in all micrographs (Figure 13a–d). While fatigue fracture zones are not prominent, root-shaped fractures are observed (Figure 13a–c). Due to the larger particle size, material removal results in greater volume and mass loss compared to nanocomposite coatings. Consequently, this leads to decreased incubation times, although the Mass Density Effect (MDE) remains lower than that of pure resin (Table 1). The short incubation time makes it difficult to use for the purpose of protecting metals from the cavitation process.

Keep in mind that poxy resins are currently utilized for protection against cavitation erosion, for example on ship blades or propellers. There are even varnishes containing micrometric particles (protected by trademark patents), although specifics regarding size, quantity, or composition remain undisclosed. Therefore, based on the findings of this study, the incorporation of nanometric boron carbide at a proportion of 6 wt.% is proposed as a reinforcement in epoxy resin formulations intended for varnishes or paints. These coatings could be applied to turbines or ship blades, offering enhanced resistance to cavitation and corrosion protection. Such varnishes or paints could facilitate rapid repairs in the industry while providing superior cavitation resistance and metal preservation.

## 4. Conclusions

In this study, composite materials were fabricated using nanoparticles, microparticles, and a commercially available epoxy resin. The extrusion mixing fabrication method achieved a homogeneous mixture, although the subsequent deaeration was inadequate for the nanocomposites to completely eliminate air bubbles. This was attributed to the formation of the surface skin, where surface curing occurred faster than in the interior due to the increased viscosity of the mixture.

Regarding the wear tests, both bulk specimens and coatings were evaluated. Generally, the coefficient of friction (FC) is higher for the coatings compared to the bulk material, except for E12SCn and E12BCn where the opposite trend was observed. This discrepancy may be attributed to the large volume of added particles, which could form agglomerates that detach more easily during wear due to inadequate anchoring with the matrix or support. Additionally, this observation aligns with the higher FC observed in the case of SiC.

However, all coatings displayed similar FC values, ranging between 0.45 and 0.5. In contrast, bulk samples exhibited comparable FC values (between 0.4 and 0.5), except for E12SCn with a FC of 0.7, and the nanocomposites of E6SCn and E6BCn, which displayed an FC of 0.3.

The wear mechanism appears to be abrasive-adhesive, involving material pull-out and the formation of wedge-like agglomerates. The percentage of filler, its size, and its anchorage with the matrix, along with proper mixing, are critical factors for increasing wear resistance. Increasing the filler percentage may lead to agglomerates of nanoparticles within the composite material, leaving more free volume upon detachment, favoring the adhesive mechanism.

As particle size increases, abrasive wear is promoted due to the angular shape of the particles, and the formation of wedge-like agglomerates may increase, thereby promoting the adhesive mechanism or releasing as debris dust, thus increasing wear. Matrix–particle anchorage enhances mechanical properties in general, particularly wear resistance.

In cavitation erosion, factors such as porosity and anchorage between particles and the matrix decrease erosion resistance. Porosity in nanocomposite coatings favored cavitation in this study. However, the behavior of nanocomposite coatings is superior to that of microcomposites and pure resin. Cavitation occurred around these pores or bubbles, serving as crack nucleation sites. This effect slightly reduces incubation time compared to pure resin. Nevertheless, the presence of larger root-shaped fracture zones, a slower form of fatigue fracture than brittle fracture, along with delamination and ductile fracture zones, results in lower MDE compared to pure resin and microcomposite coatings.

Conversely, microcomposite coatings predominantly exhibit brittle fractures and small root-shaped fracture zones, leading to higher material removal and spalling, thereby increasing MDE and reducing incubation time.

Nanocomposites with 6% fillers demonstrated better behavior than those with 12% fillers. Among E6BC and E6SC, E6BC exhibited superior properties due to better anchoring with the matrix, due to hydroxyl groups on their surface originating from the manufacturing process.

The incorporation of nanometric boron carbide at a 6% proportion in epoxy resins is proposed as a promising solution to increase resistance to cavitation and corrosion in applications such as coatings for turbines or ship blades. This coating could offer superior protection and enable rapid repairs in the industry.

## Figures and Tables

**Figure 1 polymers-16-00878-f001:**
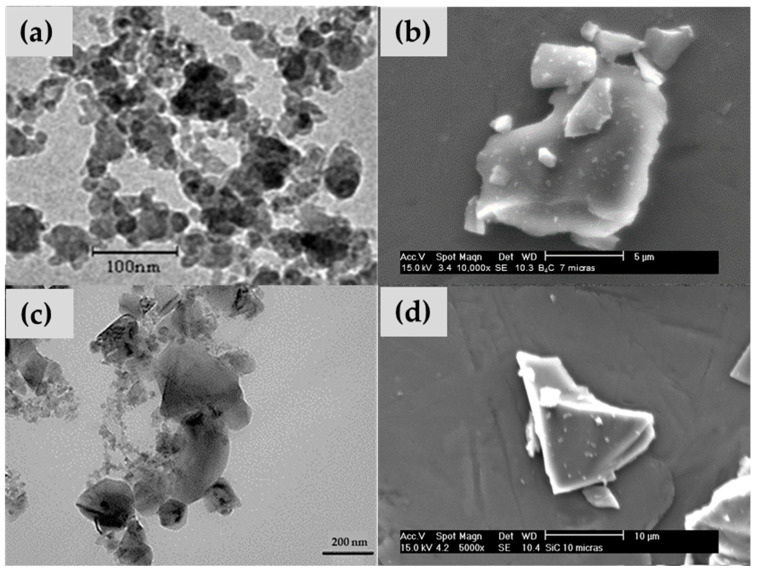
(**a**) Nanometric particles, and (**b**) micrometric particles of boron carbide, (**c**) nanometric particles, and (**d**) micrometric particles of silicon carbide.

**Figure 2 polymers-16-00878-f002:**
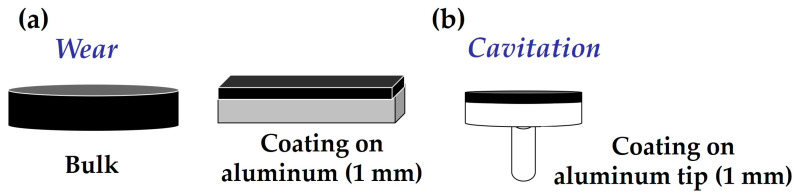
(**a**) Schematic of bulk and coating wear samples, (**b**) schematic of cavitation samples.

**Figure 3 polymers-16-00878-f003:**
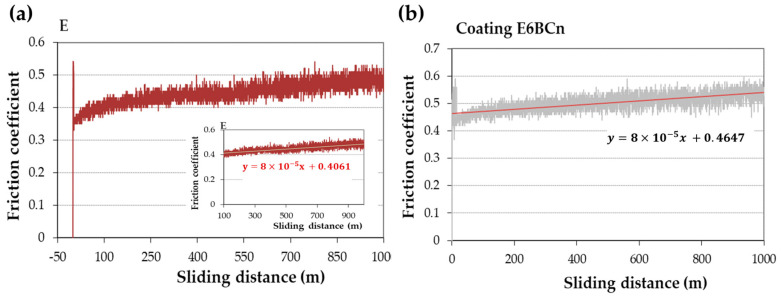
Examples of friction coefficient curves for (**a**) bulk pure resin and (**b**) coating of nanocomposite E6BCn on aluminum.

**Figure 4 polymers-16-00878-f004:**
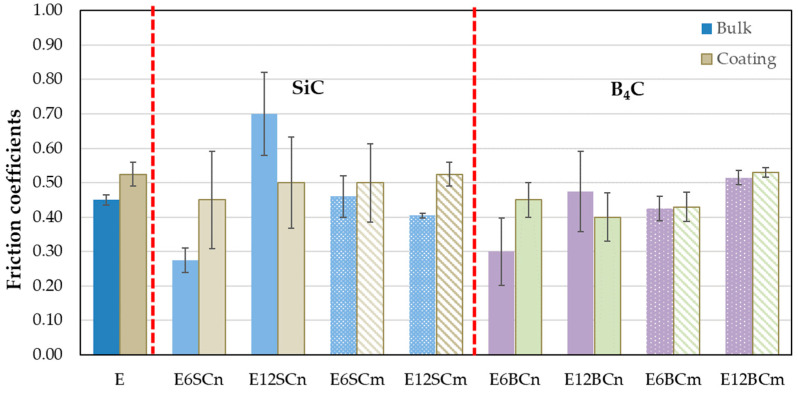
Friction coefficients for pure resin, nano- and microcomposites in bulk, and coatings on aluminum.

**Figure 5 polymers-16-00878-f005:**
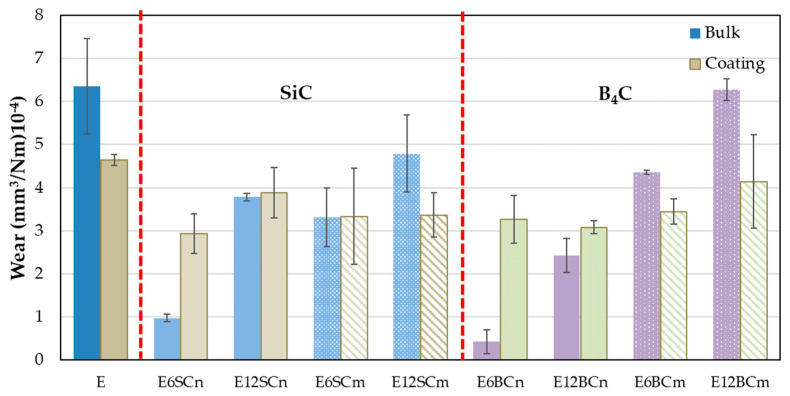
Wear for pure resin, nano- and microcomposites in bulk, and coatings on aluminum.

**Figure 6 polymers-16-00878-f006:**
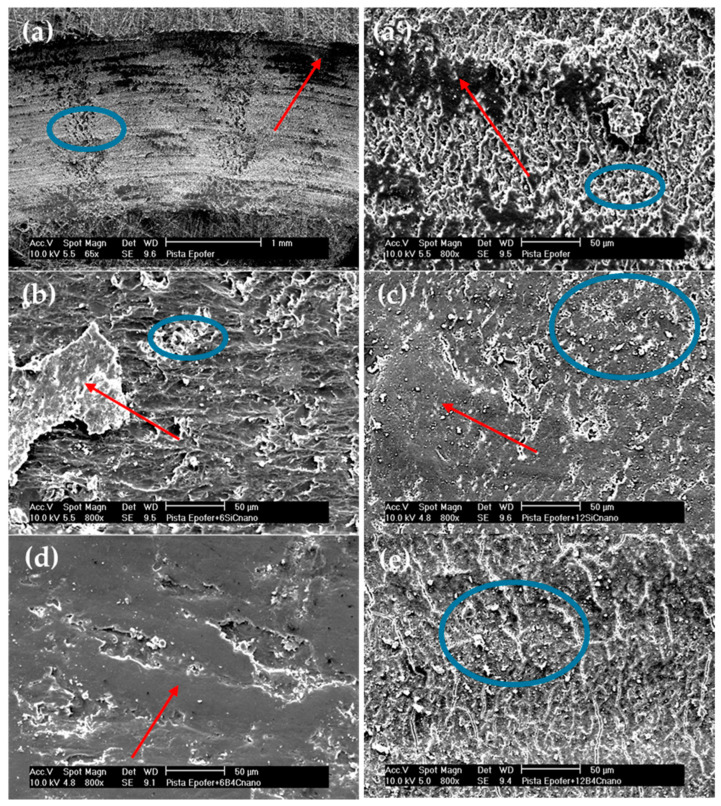
Micrographs of wear tracks: (**a**,**a’**) pure resin, (**b**) E6SCn, (**c**) E12SCn, (**d**) E6BCn, and (**e**) E12BCn. Red arrows mark areas of agglomerates or adhesion deposits and blue circles mark areas of abrasion and fatigue.

**Figure 7 polymers-16-00878-f007:**
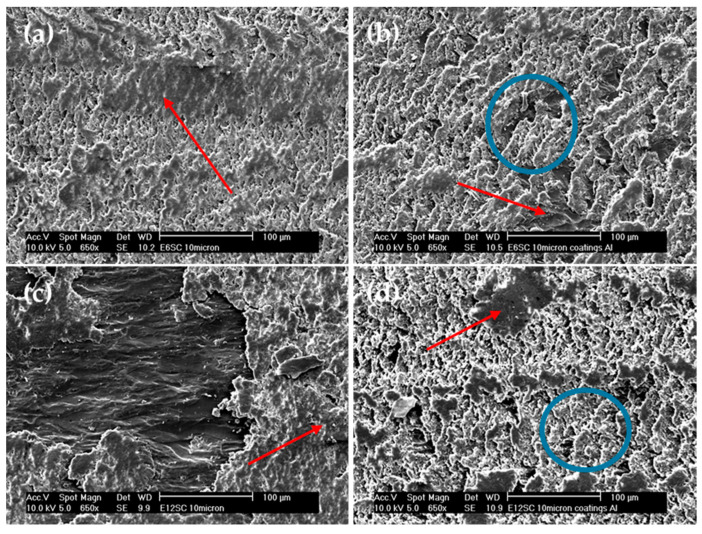
Micrographs of wear tracks. Comparison between bulk samples and coatings for microcomposites of SiC: (**a**) bulk E6SCm, (**b**) coating E6SCm, (**c**) bulk E12SCm, and (**d**) coating E12SCm. Red arrows mark areas of agglomerates or adhesion deposits and blue circles mark areas of abrasion and fatigue.

**Figure 8 polymers-16-00878-f008:**
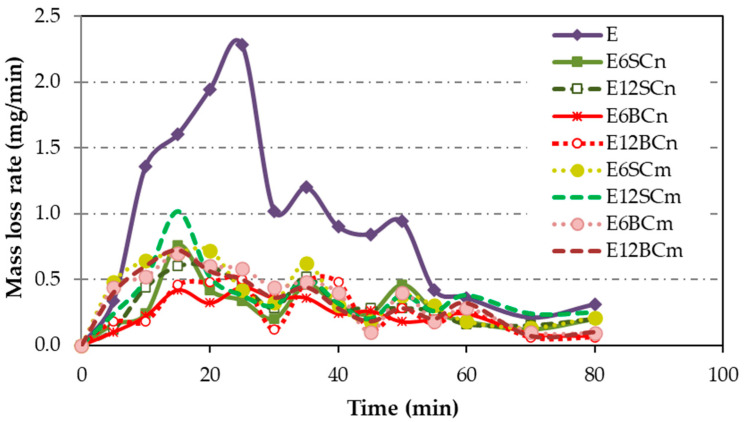
The mass loss rate for coatings of pure resin, nano- and microcomposites.

**Figure 9 polymers-16-00878-f009:**
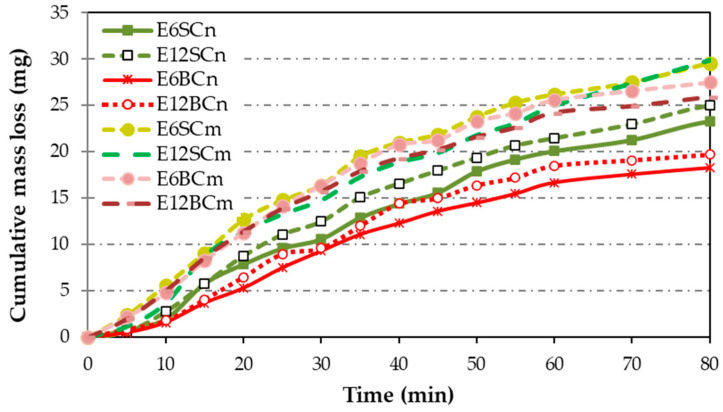
Cumulative mass loss for coatings of nano- and microcomposites. The E curve was omitted to see the differences between nanocomposite and microcomposite coatings.

**Figure 10 polymers-16-00878-f010:**
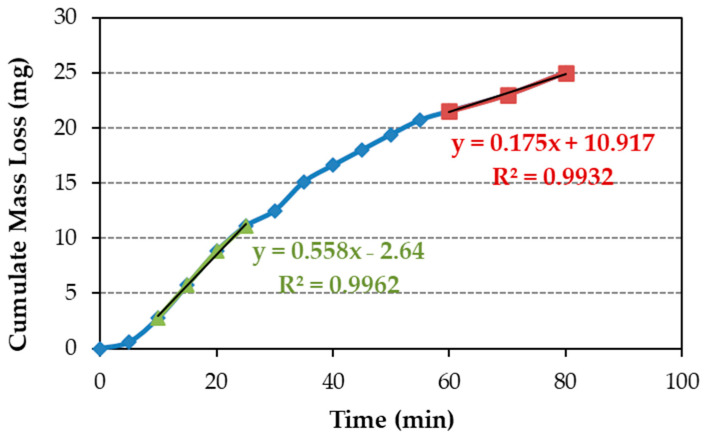
Cumulative mass loss for coatings of E12SCn (line blue with blue rhombuses). The area where the maximum erosion rate is calculated is marked with the green line with green triangles, and the area to calculate the terminal erosion rate is marked with the red line with red squares.

**Figure 11 polymers-16-00878-f011:**
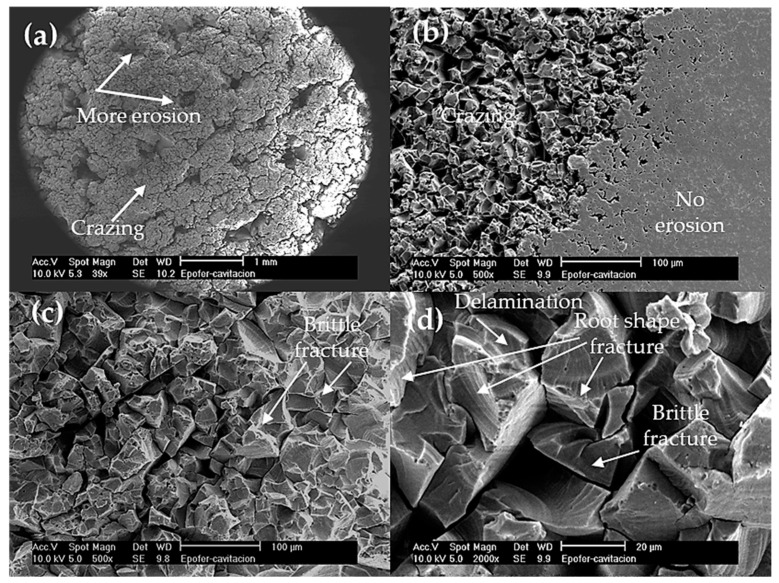
SEM micrographs. Aspects of the cavitation surface of the pure resin coating (E). (**a**) General view of the cavitation surface, (**b**) detail of the separation between the zone where erosion occurs and the zone at the edge of the sample without erosion, (**c**) zone with cavitation erosion, and (**d**) detailed zone eroded at higher magnification to observe the brittle fracture.

**Figure 12 polymers-16-00878-f012:**
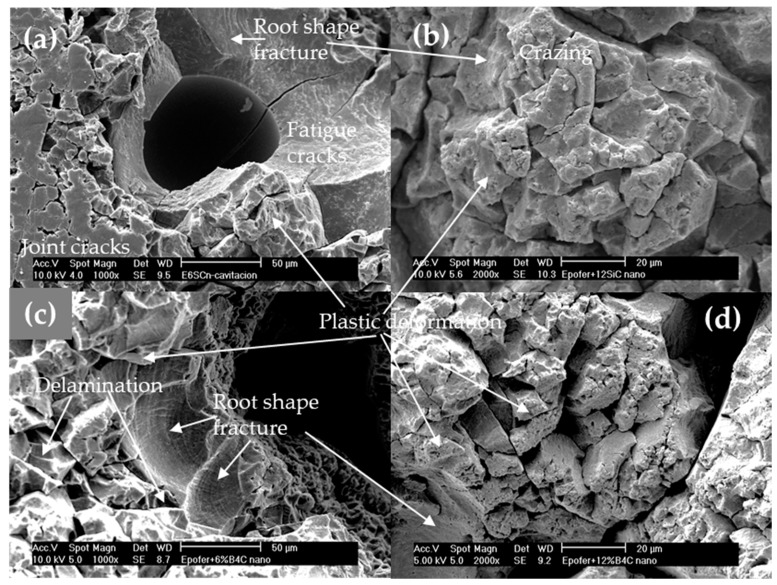
SEM micrographs. Aspects of the cavitation surface of the nanocomposites coatings: (**a**) E6SCn, (**b**) E12SCn, (**c**) E6BCn, and (**d**) E12BCn.

**Figure 13 polymers-16-00878-f013:**
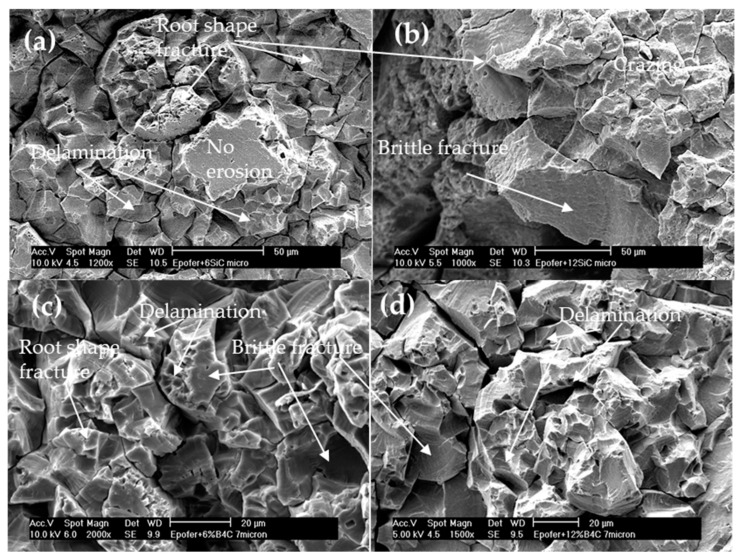
SEM micrographs. Aspects of the cavitation surface of the microcomposite coatings: (**a**) E6SCm, (**b**) E12SCm, (**c**) E6BCm, and (**d**) E12BCm.

**Table 1 polymers-16-00878-t001:** Maximum and terminal erosion rate, incubation time and MDE for pure resin and nano- and microcomposite coatings.

Materials	Maximum Erosion Rate (mg/min)	Terminal Erosion Rate (mg/min)	Incubation Time (min)	MDE at 80 min (µm)
E	2.11	0.26	7.3	240
E6SCn	0.59	0.16	6.1	76.0
E12SCn	0.56	0.18	4.7	80.8
E6BCn	0.39	0.08	5.8	59.9
E12BCn	0.48	0.06	6.4	63.1
E6SCm	0.71	0.17	2.1	94.3
E12SCm	0.45	0.25	2.9	92.0
E6BCm	0.54	0.10	0.6	88.4
E12BCm	0.53	0.09	1.3	80.4

## Data Availability

The excel files used in this work, or the micrographs produced, are available for consultation by contacting the corresponding author.

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
