# Peer review of "Wear Behavior of Epoxy Resin Reinforced with Ceramic Nano- and Microparticles"

_polymers, 2024, doi:10.3390/polym16070878_

Round 1
Reviewer 1 Report
Comments and Suggestions for Authors
This article analyses how the addition of ceramic nano-particles of SiC and B4C reinforcing epoxy matrices and their influence on mechanical erosion caused by cavitation. Nano and micro scale particles of silicon carbide (SiC) or boron carbide (B4C) are added as reinforcement at 6 wt% and 12 wt% ratios. Wear tests are conducted on bulk samples and 1mm coated aluminum sheets samples. The work is justified and of great interest to various industrial processes.
The introduction is systematized with a good bibliographic review base. The methodologies of characterization was described step by step. The results are described sequentially, using figures that promotes a good understanding of the results. The figures are taken care of and the best results are highlighted. The discussion of the results and, consequently, the conclusions, is carried out with the help of comparing the results and using some literature.
However, reading the article left me with a few questions that may help the authors to improve the discussion, such as:
In introduction the importance of studying cavitation is well justified. However, the third paragraph (lines 53-61) is confusing and should be revised. I don't understand why the advantages of numerical techniques are mentioned in this particular work. Furthermore, the use of wear tests does not seem to me to be properly justified.
Line 138 - what are the dimensions of the bulk sample?
Line 141 – “were initially sanded roughly…” Using a preferential direction or in several directions at random?
Line 157 – “The coating was applied to aluminum, as shown in Figure 2a.” In my opinion, this sentence is the same as the one in line 138.
Line 174 – “SEM is widely used by tribology researchers”. It seems an obvious phrase to me!
Line 176 – Revise the phrase: ASTM is repeated.
Line 186 – “using a sputtering system”. The equipment must be mentioned.
Line 194 – “the first part of the curve”. Are the time intervals the same for all samples? What are they? This influences the results in figure 10 and table 1. Are the time intervals the same?
Line 287 – Figure 6 - between the transition region and the centre of the tracks there are differences in compaction, agglomeration and/or volume of debris particles and abrasion mechanisms?
Line 378 – the delamination mechanism in the samples is not explained. In my opinion, delamination should be explained with the help of a figure.
Line 453 – “The manufacturing of this nano-composite (E6BCn) as paints could potentially allow for rapid industrial repairs”. In my opinion, the phrase is too general. The potential applications should be more detailed in order to understand the advantages of these coatings.
Reviewer 2 Report
Comments and Suggestions for Authors
Dear Authors.
The paper titled “Wear behavior of epoxy resin reinforced with ceramics nano and micro-particles" presents original and interesting results. However, the paper needs to be revised to improve the quality of presentation and understanding based on the following comments:
1. Please change "are" to be "were", because the tests were conducted in the past. Please, check other part of the paper that has the same condition. [line 18]
2. Please add 1 or 2 more references to support this statement [line 53-55].
3. Please add one or two more references to support this statement. [line 72-73]
4. Please describe more about why epoxy resin was selected as a substrate in this study, as it is related to the application of this study to increase erosion resistance towards cavitation, such as at the propeller, impellers, etc. Please refer to the references that epoxy is also applied as an impeller, propeller, or other components that might endure cavitation. [line 79-80]
5. Please explain more about the reason for selecting ceramic nanoparticles of Silicon Carbide and Boron Carbide to be incorporated into epoxy matrices. [line 101-102].
6. Please explain more about the meaning of comparison with the effect of micro-particles. What material of micro-particles do authors mean? [line 108].
7. What is the bulk material used in this study? [line 138]
8. Please explain why aluminium was selected as a substrate in this study [line 139].
9. Please add a caption for each figure and explain in the paragraph [line 150].
10. Please insert this standard in the references section. Please also explain how to calculate the coefficient of friction [line 159].
11. All variables should be written in italics. Please check for other variables. [line 162]
12. Please write the setting of parameters used during the cavitation-erosion test, such as frequency, etc. [line 176].
13. This ASTM standard should be written in the references section. [line 176]
14. Is the statement from other references or based on the result? If it is based on other references please mention it and write in the references section [line 217].
15. Please add a caption in each figure. [line 223].
16. Please explain more clearly why Frictional coefficient for bulk is low but the wear is higher than other specimens. [line 242]
17. Please mark where wedge-like agglomerates exist in the SEM images. [line 269]
18. Please explain more why the data has trends like this where micro ceramic coating has higher cumulative mass loss than nano-ceramic coating. [line 319]
19. Please explain more about this conclusion as the foundation for writing this conclusion. Why did E6BC exhibit superior properties due to better anchoring with the matrix by discussing and mentioning in the related SEM images? [line 450].
20. The citation is not the same the original paper. Please also add doi number. [line 476]
Please check other references.
21. Please revise the paper based on the above comments and in the paper. Please highlight the revised part with green color and to be reviewed again.
Kind regards,

Dear authors,
The quality of the English language in the paper is good and can be understood well. There are a few minor typing mistakes.
Round 2
Reviewer 2 Report
Comments and Suggestions for Authors
Dear Authors.
Thank you for the author’s revision on the paper titled “Wear behavior of epoxy resin reinforced with ceramics nano and micro-particles". However, the paper needs minor revisions to improve the quality of the paper as follows:
1. “bisphenol A” should be revised to be “bisphenol A”. [line 80]
2. “Figure 5..” should be revised to be “Figure 5.” . [line 268]
3. Please check other mistyping errors if any.
The paper has been revised according to my comments, there are only minor revisions. After the paper is revised by the authors, I will accept the revised paper.
Kind regards,

Dear authors,
The quality of the English language in the paper is good and can be understood well. There are a few minor typing mistakes.